# Lactate Levels in a Replanted Limb as an Early Biomarker for Assessing Post-Surgical Evolution: A Case Report

**DOI:** 10.3390/diagnostics15060688

**Published:** 2025-03-11

**Authors:** Alina Belu, Viorel Țarcă, Nina Filip, Elena Țarcă, Laura Mihaela Trandafir, Rodica Elena Heredea, Silviana Chifan, Diana Elena Parteni, Jana Bernic, Elena Cojocaru

**Affiliations:** 1Department of Morphofunctional Sciences I—Pathology, “Grigore T. Popa” University of Medicine and Pharmacy, 700115 Iasi, Romania; belu_alina@d.umfiasi.ro (A.B.); parteni.diana-elena@d.umfiasi.ro (D.E.P.); elena2.cojocaru@umfiasi.ro (E.C.); 2Department of Preclinical Disciplines, Faculty of Medicine, Apollonia University, Strada Păcurari nr. 11, 700511 Iași, Romania; vtarca@gmail.com; 3Department of Morphofunctional Sciences II—Biochemistry, “Grigore T. Popa” University of Medicine and Pharmacy, 700115 Iaşi, Romania; nina.zamosteanu@umfiasi.ro; 4Department of Surgery II—Pediatric Surgery, “Grigore T. Popa” University of Medicine and Pharmacy, 700115 Iasi, Romania; 5Department of Mother and Child—Pediatrics, “Grigore T. Popa” University of Medicine and Pharmacy, 700115 Iasi, Romania; laura.trandafir@umfiasi.ro; 6Department I Nursing, Discipline of Clinical Practical Skills, “Victor Babeş” University of Medicine and Pharmacy, 300041 Timişoara, Romania; elena-rodica.heredea@umft.ro; 7Faculty of Medicine, “Grigore T. Popa” University of Medicine and Pharmacy, 700115 Iasi, Romania; silviana.chifan@yahoo.com; 8Discipline of Pediatric Surgery, “Nicolae Testemițanu” State University of Medicine and Pharmacy, MD-2001 Chisinau, Moldova; jana.bernic@usmf.md

**Keywords:** lactate, pediatric trauma, limb replantation, postoperative prognosis

## Abstract

**Background and Clinical Significance:** In the clinical management of major pediatric traumatic injuries and other hypoxic conditions, lactate is widely recognized as a key indicator of tissue hypoxia and potential necrosis. However, its prognostic value remains uncertain. Several factors influence post-surgical outcomes, including the time between amputation and replantation, transport conditions, asepsis, the extent of tissue necrosis, hemorrhagic shock, coagulation disorders, and the heightened risk of contamination. **Case presentation:** We present this case to emphasize the utility of systemic lactate versus lactate levels in the replanted limb for monitoring post-transplantation outcomes in a pediatric patient with traumatic limb amputation. Significant fluctuations in lactate levels within the replanted limb were observed at the onset of unfavorable evolution, specifically on the seventh postoperative day, coinciding with the identification of *Aspergillus* spp. infection. This necessitated the use of synthetic saphenous vein grafts and Amphotericin B administration. Despite these interventions, disease progression ultimately led to limb amputation. **Conclusions:** Lactate levels in the replanted limb may serve as an early biomarker for assessing post-surgical evolution. However, further case reports are required to confirm its predictive value.

## 1. Introduction

Several key factors must be considered when evaluating a replantation strategy for an amputated limb. While early replantation is the preferred approach for traumatic limb amputation, it is not always feasible or successful. The decision between limb salvage and amputation should prioritize post-surgical functional outcomes and overall survival benefits [1,2]. The post-surgical prognosis is influenced by several factors, including the time elapsed between amputation and replantation, transport conditions and asepsis of the amputated limb, the extent of tissue necrosis (indicated by lactate levels), hemorrhagic shock (assessed through hemoglobin levels), coagulation disorders (such as thrombosis risk), and the heightened risk of contamination (e.g., with *Aspergillus* spp.) [3,4,5]. In some patients undergoing replantation, hemorrhagic shock accompanied by acute hypoxia and hypotension, as well as sepsis, can lead to lactic acidosis, thereby worsening the surgical prognosis [6]. Previous studies have shown that low lactate clearance within the first 24–48 h and elevated lactate levels in patients with sepsis or septic shock are associated with increased morbidity and mortality. Consequently, lactic acid levels may serve as a valuable non-invasive parameter for monitoring and managing sepsis [5,7].

Lactate is primarily an energy-yielding, non-toxic substrate. Under resting conditions, approximately half of the total lactate produced (1500 mmol/day) is utilized for gluconeogenesis in the liver via the Cori cycle, while the remaining 50% undergoes oxidation [6]. At the cellular level, adrenergic stress in shock patients accelerates glycolysis, leading to increased lactate production [8]. Skeletal muscle is a major contributor to lactate formation, as excessive aerobic glycolysis stimulates Na^+^/K^+^ ATPase activity during septic shock [8,9].

Lactic acid serves as a key metabolic parameter for monitoring the postoperative progression of limb replantation. In cases of delayed replantation following traumatic amputation, lactate levels in the replanted limb provide a more precise indicator of recovery compared to systemic lactate, which is influenced by metabolic compensation mechanisms involving the liver, kidneys, and respiratory system [10].

Although various current protocols [11,12,13,14,15] recommend the initiation of antimicrobial therapy, necrosis may still occur in rare cases due to uncommon microorganisms. There are very few studies exploring lactate monitoring as a risk factor for limb replantation failure, and even fewer in pediatric patients, where no standardized protocol has been established.

This study aims to emphasize the utility of monitoring lactate levels in replanted limbs over systemic lactate values as a more reliable indicator of postoperative evolution.

## 2. Case Description

### 2.1. Signs, Symptoms, and Management

A 6-year-old girl involved in a car accident sustained multiple traumatic injuries and was admitted to the “St. Mary” Emergency Clinical Hospital for Children in Iași, Romania, approximately seven hours post-accident. She underwent limb replantation following a traumatic amputation.

Upon admission, she was in critical condition, presenting with a crush amputation of the left lower limb at the proximal third of the femur, a left femur fracture, severe concussion of the right sciatic nerve, a pelvic fracture, and a crushed right lower limb with significant soft tissue loss. Additionally, she had a fractured left clavicle, fractured ribs, left lung contusion, and signs of hemorrhagic shock.

Hemodynamic evaluation at admission highlighted: Heart rate (HR) of 170 beats per minute, blood pressure (BP) of 100/60 mmHg, hemoglobin (Hb) value of 8.6 g/dL, hematocrit (Hct) of 28%, blood pH of 7.24, PaCO_2_ of 54 mmHg, PaO_2_ of 54 mmHg, systemic lactate of 1.7 mmol/L, SvO_2_ of 81%.

Surgical limb replantation was performed 12 h after the accident, including open reduction and osteosynthesis with a plate and three screws for the left femur, distal one-third replantation of the left lower limb, arteriorrhaphy using a bilateral saphenous vein graft, and phleborrhaphy with a contralateral saphenous vein graft. To prevent post-replantation edema, early prophylactic fasciotomy was carried out on all four muscle compartments of the calf as well as the dorsal and plantar compartments of the foot.

Due to hemorrhagic shock, the patient required both pre- and intraoperative transfusions. To ensure adequate limb perfusion, hemoglobin levels were maintained at 12 g/dL, in accordance with current guidelines. One week after replantation, an infection with *Aspergillus* was detected in the wound at the left thigh and shank, necessitating the replacement of the left saphenous vein grafts with synthetic ones and the administration of Amphotericin B. The patient received a loading dose of 70 mg/m^2^, followed by a daily maintenance dose of 50 mg/m^2^.

Despite these interventions, the clinical course was unfavorable, and limb amputation became necessary due to the major risk of systemic fungal dissemination, which could have led to a fatal outcome. Following closure of the left thigh stump and the right thigh wound, the patient’s condition improved.

Written informed consent was obtained from the patient’s parents for publication of this case. The study conforms to CARE guidelines [16].

### 2.2. Biochemical Monitoring

The analysis included the dynamic monitoring of several biochemical parameters over a ten-day post-traumatic period: systemic lactate measured via a central venous catheter (CVC), lactate levels in the replanted limb assessed through a femoral vein catheter, hematocrit (Hct), partial pressure of oxygen (PaO_2_), blood glucose levels, and venous oxygen saturation (SvO_2_). These parameters were determined using the GEM Premier 3500 system (Instrumentation Laboratory Co., Lexington, KY, USA).

Biochemical monitoring was conducted across three distinct time intervals: prior to replantation (7–10 h post-accident), post-replantation (10–171 h post-accident), and at the time of limb amputation (213 h post-accident). Measurements were taken at one-hour or two-hour intervals, depending on vascular access feasibility.

Currently, no standardized monitoring protocol is available in the literature for this specific context.

### 2.3. Statistical Analysis

Statistical analysis of the recorded biological parameter values was performed using SPSS 25 (IBM Corporation, New Orchard Road, Armonk, NY, USA). Due to the limited number of measurements, non-parametric tests were applied for comparisons, as the data did not follow a normal distribution. Normality was assessed using the Kolmogorov–Smirnov test.

To compare the parameters across the three analyzed intervals (before replantation, after replantation, and during massive Aspergillus invasion), the Kruskal–Wallis H test was applied. The Mann–Whitney U test was used to compare lactate levels in the limb between the post-replantation period and the stage of Aspergillus invasion.

### 2.4. Biochemical Parameter Results

#### 2.4.1. Biochemical Evaluation

Biochemical parameters monitored over the three time intervals are presented in Table 1. Figure 1 shows a comparative analysis of systemic lactate values versus lactate values in replanted limb.

#### 2.4.2. Lactate Level

According to the data in Table 1, lactate levels in the replanted limb did not match systemic lactate levels at the time of limb revascularization (12.7 mmol/L vs. 5.7 mmol/L) (Figure 2). However, lactate concentrations in the replanted limb were significantly higher than systemic values following replantation, likely due to the reperfusion of ischemic tissue.

Subsequent wound measurements showed a progressive decrease in lactate levels, indicating active perfusion of the replanted limb until the onset of *Aspergillus* infection. At that point, lactate levels in the limb rose to 12.2 mmol/L, in contrast to the systemic lactate value of 1.8 mmol/L (Figure 2).

Prior to replantation, systemic lactate levels were relatively normal. After replantation, systemic lactate increased by four units but normalized within approximately ten hours. In the following days, both systemic lactate and lactate levels in the replanted limb remained normal. However, on the seventh day, *Aspergillus* infection was detected in the wound. At that point, the venous grafts were replaced with synthetic ePTFE grafts, and treatment with Amphotericin B was initiated.

Over the next two days, systemic lactate levels were continuously monitored. While systemic lactate remained within normal limits, lactate in the wound exhibited an exponential increase. Notably, the patient did not receive any blood transfusions after replantation.

To compare the dynamic evolution of the three critical intervals, we presented the average values of systemic lactate and lactate in the replanted limb. The mean systemic lactate value was normal at the time of *Aspergillus* infection, but it differed significantly from the lactate level in the replanted limb, which was elevated at the time of infection (Figure 1b).

#### 2.4.3. Partial Pressure of Oxygen (PaO_2_)

PaO_2_ values remained within normal range, indicating that in this polytrauma patient, pulmonary oxygenation was effective and that oxygen delivery to the replanted limb was optimal (Table 1).

#### 2.4.4. Venous Oxygen Saturation (SvO_2_)

Increased SvO_2_ values, exceeding the upper limits, indicate inefficient oxygen consumption by the replanted limb, suggesting a degree of ischemia. Following limb replantation, SvO_2_ values initially decreased, reflecting oxygen consumption at the tissue level, but then increased again until Aspergillus infection was detected. The elevated SvO_2_ values were linked to high lactate levels in the limb, while systemic lactate remained within normal limits. There was a positive correlation between elevated SvO_2_ values and high lactate levels in the limb, indicating tissue death (Table 1).

#### 2.4.5. Methods Section Describing the Histological Analysis

The collected tissue samples were fixed in 10% buffered formalin for 24–48 h, followed by standard histological processing, including dehydration in increasing concentrations of ethanol, clearing in xylene, and embedding in paraffin. Sections of 3–5 µm thickness were obtained using a microtome and mounted on glass slides. For routine histological analysis, the sections were stained with hematoxylin and eosin (H&E) using a standard protocol to highlight tissue architecture and cellular components. Additionally, to identify fungal elements, the sections underwent Periodic Acid-Schiff (PAS) staining. After treatment with periodic acid, the sections were incubated with Schiff reagent, followed by counterstaining with Mayer’s hematoxylin to highlight cell nuclei. Microscopic examination was performed using a light microscope Nikon Eclipse E600, Boston Industries, Inc., Walpole, Massachusetts, USA and images were captured using a digital imaging system.

Tissue samples infected with *Aspergillus* spp. were processed using the paraffin inclusion technique and examined with hematoxylin and eosin (HE) staining, as well as Periodic Acid-Schiff (PAS) staining. Histological analysis revealed striated muscle fibers with necrosis, significant polymorphous inflammatory infiltration predominantly consisting of neutrophils, hemorrhagic areas (Figure 3), and the presence of fungal elements.

The special PAS staining revealed 5–10 mm thick, septate, acute-angle (45°) or dichotomous branching hyphae characteristic for *Aspergillus* (Figure 4).

## 3. Discussion

Recent advances in reconstructive surgery for the reimplantation of amputated limbs have enabled a significant percentage of patients to regain limb function. However, there are still cases that necessitate surgical amputation and subsequent prosthetic use [17,18]. The main factors leading to surgical amputation include the prolonged interval between the traumatic event and surgery (in pediatrics, there is no established protocol regarding the exact timeframe for replantation after traumatic amputation) [19,20,21], transport conditions for the amputated limb, asepsis, the degree of tissue necrosis (indicated by lactate levels in the amputated limb), hemorrhagic shock (assessed by hemoglobin levels), coagulation disorders (such as the risk of thrombosis), and a high risk of infection or contamination (often with *Candida albicans* and less frequently with *Aspergillus*) [22,23].

The surgeon faces a challenging task in monitoring the postoperative evolution of replanted limbs. The biochemical parameters used to assess revascularization during limb replantation include hematocrit (Hct), blood glucose (glycemia), partial pressure of oxygen (PaO_2_), venous oxygen saturation (SvO_2_), systemic lactate, and wound lactate [24,25]. Among these, lactate levels in the wound proved to be the most significant indicator in monitoring our patient’s progress.

The metabolic response to trauma leads to oxygen deficiency, hypoxia, and anaerobic metabolism, with lactate being the final product [26]. Lactate has long been used as a surrogate marker for tissue perfusion, and its levels have well established clinical applications in the initial assessment of polytrauma patients [27,28]. While there are several biochemical markers for acidosis, lactate levels are the most commonly utilized serum marker [29]. Lactic acidosis (defined as a serum lactate level of 4 mmol/L) occurs in trauma patients due to metabolic disturbances caused by hypoxia, hemorrhage, and anaerobic metabolism [30,31]. Polytrauma involving crushed tissues and significant mass loss inevitably results in elevated lactate levels, which can reach a fatal maximum reported in the literature as any value above 9 mmol/L, representing the threshold for cell death.

The predictive value of initial serum lactate in traumatized adult patients was explored in several studies [32]. Research by Fu, Y. et al. demonstrated that elevated serum lactate levels upon admission correlate with high serum glucose, coagulopathy, anemia, and the severity of traumatic brain injury in children [33]. However, while serum lactate is a reliable diagnostic marker for hypoxia in adults, its application in children is not fully validated due to incomplete understanding of how lactate production differs between children and adults following severe trauma [29,30].

On the other hand, serum lactate is also utilized in post-surgical follow-up and replantation to monitor the effectiveness of treatment and provide prognostic information after replantation. Studies conducted in adults have shown that lactate clearance should be considered when interpreting serum lactate values as a prognostic marker after surgery [34,35,36].

The role of systemic lactate in monitoring the replanted limb was explored in previous research. In our patient, we compared the systemic lactate values with lactate levels from the replanted limb, which primarily serve as a marker for tissue perfusion. It is understood that serum lactate levels are influenced by both tissue perfusion and lactate clearance in the liver and kidneys. In our case, the high lactate value in the replanted limb after 12 h of severe ischemia (12.7 mmol/L) was associated with a relatively modest increase of only 5.7 mmol/L in systemic lactate. Generally, the reported mortality rate for shock is 73% if systemic lactate is between 4.5 and 8.9 mmol/L, and 100% if it exceeds 13 mmol/L [35].

After resuscitation from post-hemorrhagic shock and successful limb replantation with blood reperfusion, the serum lactate levels normalized much more rapidly compared to the lactate levels in the wound, which served as a marker for revascularization and tissue oxygenation. A notable advantage in pediatric patients is that, in general, children do not suffer from associated hepatic, renal, or cardiovascular comorbidities [37,38,39]. Following this, both systemic lactate and wound lactate values were monitored simultaneously. It was observed that after one week, both systemic lactate and wound lactate levels began to rise, with systemic lactate increasing to 2.3 mmol/L, while wound lactate reached 12.2 mmol/L. This marked the moment when Aspergillus infection was diagnosed, and the right saphenous vein grafts were replaced with synthetic grafts. Additionally, Amphotericin B treatment was initiated. Severe hypoxia and tissue necrosis caused by the Aspergillus infection, without circulatory failure due to the synthetic graft, were associated with the persistence of high lactate levels in the wound. Meanwhile, serum lactate levels normalized due to the clearance mechanisms of the liver, kidneys, and respiratory system [40,41,42].

After the replanted leg was amputated, the serum lactate level normalized in our case, despite the elevated lactate level in the wound. Even though blood supply to the leg was adequate, the increase in lactate indicated that sepsis had progressed from a metabolically compensated to a decompensated stage, where tissues were using less oxygen. Overall, the utility of serum lactate as a predictive biomarker for post-surgical evolution of the replanted limb is limited. In our case, monitoring lactate levels in the replanted limb proved to be a more reliable indicator of prognosis following late limb replantation after severe amputation.

Compared to other parameters, changes in Hct were not relevant in lactate variations. The primary factor influencing lactate variations, especially in the replanted limb, was the variation in circulating volume. Systemic Hct is not a good indicator of better oxygenation in the limb. The acid-base balance in the limb would be useful to show the degree of tissue oxygenation, otherwise, it is irrelevant. Lactate does not increase systemically or in the limb if the hemoglobin (Hb) value is below 8.7 g/dL or Hct is below 28%. With a Hb value higher than 8.7 g/dL, lactate concentration in the limb is exclusively influenced by the permeability of the anastomoses and the tissues’ resistance to blood flow (e.g., compression edema, which was prevented by early prophylactic fasciotomy in this case). Notably, the replanted limb showed favorable progress until the *Aspergillus* infection developed. This suggests that even delayed replantation (more than 10 h after traumatic injury) can have high success rates in children, provided a strict protocol is followed to prevent infection by rare and virulent pathogens. Due to the risk of sepsis, which could have been fatal, surgical amputation of the replanted leg was required.

Bacterial graft infection is a rare complication, occurring in less than 3% of cases, with Candida being the predominant pathogen. *Aspergillus* infections in vascular grafts are more commonly reported in aortic surgeries [43], though they are extremely rare in infrainguinal prosthetic graft infections, typically occurring during graft placement via airborne fungal spores [23,44,45,46,47,48]. Despite being rare, Aspergillus infections in vascular grafts are associated with a very high mortality rate (50–95%) due to diagnostic challenges, limited antifungal treatment options, frail patient populations, and a poor understanding of the virulence factors that contribute to Aspergillus pathogenicity and its interactions with the host immune system [49,50,51,52,53].

In our case, lactate levels in the replanted limb were monitored as part of the post-surgical evolution. The decision to amputate the limb was made due to the risk of systemic dissemination, which also correlated with the increase in lactate levels. Lactate serves as a valuable marker because it is inexpensive, widely available, and provides quick feedback on the effectiveness of interventions. Therefore, it can be a useful tool for monitoring polytrauma therapy or systemic infections [54].

This case was informed by our prior experience using lactate measurements in the replanted limb of several pediatric patients with major trauma. At an Hb value higher than 8.7 g/dL, lactate levels in the replanted limb were primarily influenced by the permeability of the anastomoses and the tissues’ resistance to blood flow. The adverse outcome leading to amputation was primarily due to the Aspergillus infection. Given the lack of comprehensive literature on this topic, there is a clear need for the development of a specific pediatric protocol, which should include both monitoring lactate levels in the replanted limb and a therapeutic approach to prevent infection with aggressive pathogens. Further research is needed to determine the prognostic value of tissue reperfusion and to establish lactate in the replanted limb as a biochemical marker in the prognosis of children with major trauma.

*Limitation of the study:* Our manuscript outlines an analytical approach that should be applied to a future cohort. The objective was to highlight the importance of monitoring lactate levels in the replanted limb and to emphasize its relevance in therapeutic decision-making. To establish significant differences between systemic lactate and lactate levels in the replanted limb, a statistical analysis of the data was necessary. These findings could help refine the monitoring protocol for biochemical parameters in cases of traumatic amputations. However, the conclusions drawn from this manuscript cannot yet be generalized on a statistical level, and further research is required to validate and expand upon these results. Given the absence of clear therapeutic protocols for children, we believe it is crucial to publish our findings and raise awareness in this area.

## 4. Conclusions

The monitoring protocol for biochemical parameters in traumatic amputations highlights the crucial role of lactate values. Lactate serves as a marker of the balance between oxygen demand and availability, with changes in its levels providing an effective measure during resuscitation efforts. Monitoring lactate levels in the replanted limb has proven to be a more reliable indicator than serum lactate measurements in predicting the prognosis of late limb replantation following traumatic amputation in children. Furthermore, we aim to underscore the importance of proactive measures to prevent severe infections caused by microorganisms in these cases.

## Figures and Tables

**Figure 1 diagnostics-15-00688-f001:**
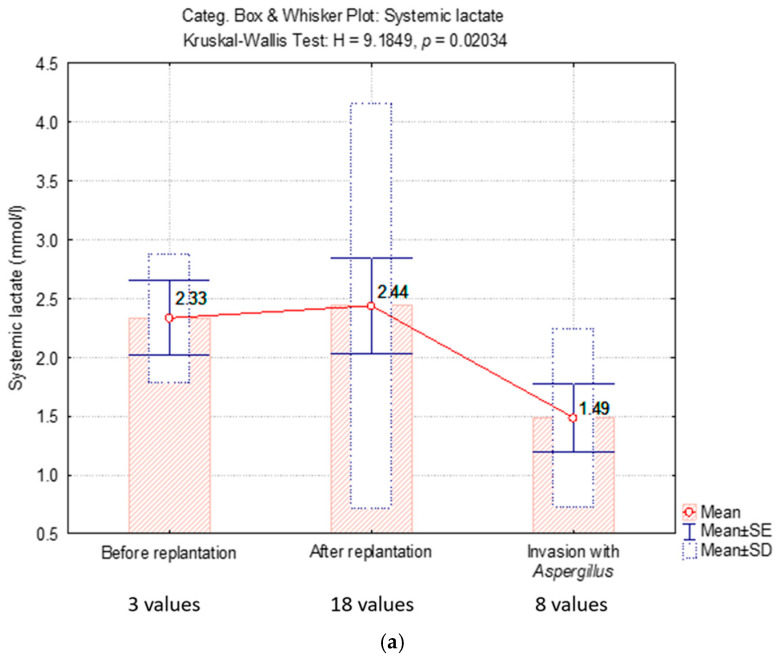
(**a**) Comparative analysis of systemic lactate values; (**b**) lactate values in replanted limb.

**Figure 2 diagnostics-15-00688-f002:**
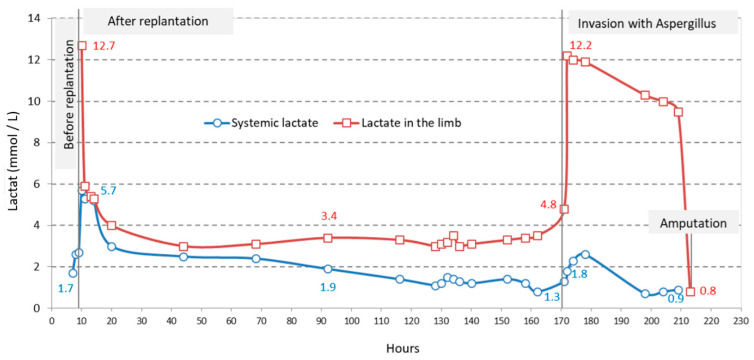
Dynamic evaluation (hours) of systemic lactate and lactate in replanted limb.

**Figure 3 diagnostics-15-00688-f003:**
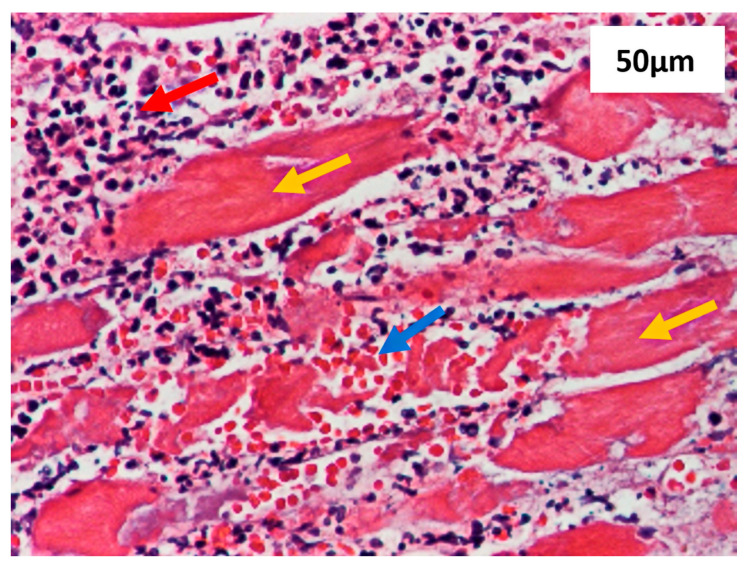
High-magnification view of muscular tissue exhibiting areas of necrosis (yellow arrows), polymorphous inflammatory infiltration (red arrow), and hemorrhage (blue arrow). Stained with hematoxylin and eosin (H&E), scale bar 50 µm, 20× magnification of camera lens.

**Figure 4 diagnostics-15-00688-f004:**
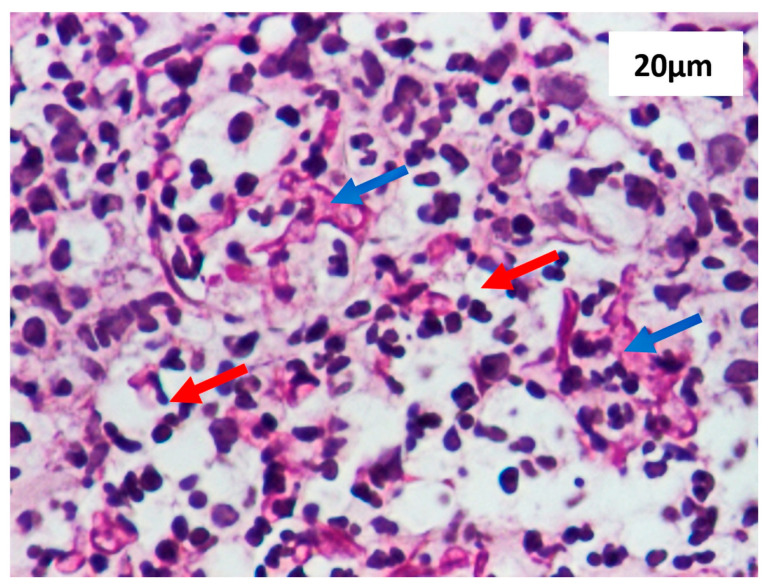
Inflammatory exudate (red arrows) with fungal elements (blue arrows), highlighted by Periodic Acid-Schiff (PAS) staining, scale bar 20 µm, 40× magnification of camera lens. In our patient, *Aspergillus* was present in wound, specifically around arteriovenous graft within anastomosis area, before insertion of synthetic graft. Following identification of *Aspergillus* spp., synthetic vascular graft was inserted to clean wound and disrupt fungal growth.

**Table 1 diagnostics-15-00688-t001:** Biochemical parameters monitored before and after surgery.

	Before Replantation (7–10 h After the Accident)	After Replantation (10–171 h After the Accident)	Massive Invasion with *Aspergillus* spp. (171–213 h After the Accident)
**Monitoring parameters**			
PaO_2_ (mmHg)			
mean ± SD	51.7 ± 2.5	42.6 ± 5.4	46.3 ± 12.6
median (Q1; Q3)	52 (49; 54)	42.5 (39; 45)	49.5 (45.5; 53.5)
Glycemia (g/dL)			
mean ± SD	169.7 ± 57.4	107.4 ± 33.5	81.1 ± 28.4
median (Q1; Q3)	177 (109; 223)	100 (95; 122)	90.5 (77.5; 98)
Hct (%)			
mean ± SD	23 ± 5	26.4 ± 5.8	29.1 ± 4.9
median (Q1; Q3)	23 (18; 28)	26 (24; 31)	31 (26.5; 32.5)
SvO_2_ (%)			
mean ± SD	79.3 ± 2.1	71.4 ± 11.5	74.4 ± 6.7
median (Q1; Q3)	80 (77; 81)	75 (62; 78)	74 (69.5; 79)
**Monitoring parameters—potential prognostic factors for the reimplanted limb**			
Systemic lactate (mmol/L)			
mean ± SD	2.3 ± 0.5	2.4 ± 1.72	1.4 ± 0.7
median (Q1; Q3)	2.6 (1.7; 2.7)	1.4 (1.2; 3)	1.3 (0.8; 2.3)
Lactate in the limb (mmol/L)	-		
mean ± SD		4.1 ± 2.3	9.6 ± 3.7
median (Q1; Q3)		3.4 (3.1; 4)	10.3 (9.5; 12)

Mean ± SD, mean ± standard deviation; Q1, lower quartile; Q3, upper quartile.

## Data Availability

The data presented in this study are available on request from the corresponding author.

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
