# Peer review of "Lactate Levels in a Replanted Limb as an Early Biomarker for Assessing Post-Surgical Evolution: A Case Report"

_diagnostics, 2025, doi:10.3390/diagnostics15060688_

Round 1
Reviewer 1 Report
Comments and Suggestions for Authors
Dear Editor and Authors,
As an orthopedic surgeon, your article really captured my attention for two main reasons:
1. Earthquakes occur quite frequently in my country.
2. The city where I live is situated on a major highway, leading to numerous traffic accidents.
Because of these factors, we often face the challenging decision of whether to amputate a replanted limb or address a crush injury. I appreciate your point about systemic lactate not being particularly helpful in this context. It is primarily useful in determining whether we should proceed with definitive treatment or opt for damage control orthopedics in multi-trauma patients.
In summary, I enjoyed your article. However, I have two suggestions for improvement:
First, I recommend changing the title, as it may mislead readers. A more fitting title could be: "Lactate Levels in a Replanted Limb as an Early Biomarker for Assessing Post-Surgical Evolution: A Case Report."
Second, I suggest including preoperative, postoperative, and pre-amputation photos of the limb to engage readers more effectively.
Kind regards.
Author Response
Dear Reviewer,
Thank you very much for evaluating our manuscript. We changed the title as you suggested.
The perioperative images are of poor quality, and we decided not to include them in the manuscript.
We provide the revised version of our manuscript; the changes we have made are highlighted in red.
Kind regards,
Reviewer 2 Report
Comments and Suggestions for Authors
I appreciate the novelty of the manuscript aiming to describe lactate levels in a replanted limb as a more reliable indicator of postoperative outcomes compared to systemic lactate levels. However, this manuscript requires major revisions to improve the scientific quality of the findings presented herein. Please see my comments and suggestions below to help improve the manuscript for peer review.
Specific comments to be addressed by authors:
- Line 148: the values referred here are shown in Figure 2, not Figure 1. The authors need to correct either the figure reference or the values referenced.
- The authors need to state the sample sizes for the three stages evaluated in Figure 1 (a) and (b). Given the variability in sampling intervals, this data is useful for the readers to interpret the data. It appears that only up to 3 measurements were taken for the “Before Replantation” group.
- Figure 1: It would be beneficial to match the scaling for both graphs to help emphasize the differences between limb lactate and systemic lactate levels.
- Figure 2: the axes need to have labels (include the title and unit of measurement).
- All figure legends require more explanation/description.
- Lines 190-192: The authors state no correlation was found between limb lactate levels and systemic lactate levels based on a Pearson correlation analysis. However they go on to state that based on this relationship, systemic lactate values are less accurate in reflecting the limb’s condition. This is an overreach in interpreting the data. All the author’s can state about this analysis is that there is no (linear) relationship between limb and systemic lactate levels during the “after replantation” phase. The authors need to revise this paragraph.
- Lines 173-175: the p-values in this statement are confusing for the reader. The authors state that the mean systemic lactate levels were normal at the time of Aspergillus infection and provide a p-value. This p-value refers to the one-way ANOVA analysis across three intervals for the systemic lactate levels, which indicates that the mean lactate levels are different from each other. The authors should remove this p-value from the sentence as it does not provide any useful information to the statement. Furthermore, the authors state that the systemic lactate levels significantly differed from the limb lactate levels and provide a p-value, but this p-value refers to a comparison between after replantation and aspergillus infection measurements from limb lactate levels. The authors need to remove this p-value as well. Rather, it would be beneficial for the authors to do a direct comparison (t-test/mann whitney) to compare the systemic and limb lactate levels at the time of infection. They should include a bar graph and description of the statistics in this section instead.
- Figure 4: the authors need to include a proper scale bar in the image with a stated unit of measurement either in the image or the figure description. It would also be helpful if the authors could annotate the images to highlight the areas of necrosis, polymorphous inflammation, and hemorrhage, specifically the fungal elements would be most useful to be pointed out.
- Figure 5: same comments as Figure 4; the authors need to include a proper scale bar in the image with a stated unit of measurement either in the image or the figure description. Furthermore, the authors should annotate the image to highlight areas of inflammatory exudate and fungal elements.
- Materials and Methods: the manuscript is completely lacking a methods section describing the histological methods used for the images shown in Figures 4 and 5. The authors need to include this section as it is part of the manuscript.
Author Response
I appreciate the novelty of the manuscript aiming to describe lactate levels in a replanted limb as a more reliable indicator of postoperative outcomes compared to systemic lactate levels. However, this manuscript requires major revisions to improve the scientific quality of the findings presented herein. Please see my comments and suggestions below to help improve the manuscript for peer review.
Response: Dear Reviewer, Thank you very much for evaluating our manuscript. Your recommendations and comments have helped us improve our manuscript. Here we provide the requested corrections and address the comments. The changes we have made in the manuscript are highlighted in red.
Specific comments to be addressed by authors:
- Line 148: the values referred here are shown in Figure 2, not Figure 1. The authors need to correct either the figure reference or the values referenced.
Response: We corrected.
- The authors need to state the sample sizes for the three stages evaluated in Figure 1 (a) and (b). Given the variability in sampling intervals, this data is useful for the readers to interpret the data. It appears that only up to 3 measurements were taken for the “Before Replantation” group.
Response: We corrected Figure 1 and added the number of measurements for each period.
- Figure 1: It would be beneficial to match the scaling for both graphs to help emphasize the differences between limb lactate and systemic lactate levels.
Response: Since the differences between the values of systemic lactate versus the replanted limb were very large, within a single graph the differences would have faded visually for systemic lactate.
- Figure 2: the axes need to have labels (include the title and unit of measurement).
Response: We corrected Figure 2 and added labels.
- All figure legends require more explanation/description.
Response: We added more explanation for legends.
- Lines 190-192: The authors state no correlation was found between limb lactate levels and systemic lactate levels based on a Pearson correlation analysis. However they go on to state that based on this relationship, systemic lactate values are less accurate in reflecting the limb’s condition. This is an overreach in interpreting the data. All the author’s can state about this analysis is that there is no (linear) relationship between limb and systemic lactate levels during the “after replantation” phase. The authors need to revise this paragraph.
Response: We deleted this paragraph. Thank you.
- Lines 173-175: the p-values in this statement are confusing for the reader. The authors state that the mean systemic lactate levels were normal at the time of Aspergillus infection and provide a p-value. This p-value refers to the one-way ANOVA analysis across three intervals for the systemic lactate levels, which indicates that the mean lactate levels are different from each other. The authors should remove this p-value from the sentence as it does not provide any useful information to the statement. Furthermore, the authors state that the systemic lactate levels significantly differed from the limb lactate levels and provide a p-value, but this p-value refers to a comparison between after replantation and aspergillus infection measurements from limb lactate levels. The authors need to remove this p-value as well. Rather, it would be beneficial for the authors to do a direct comparison (t-test/mann whitney) to compare the systemic and limb lactate levels at the time of infection. They should include a bar graph and description of the statistics in this section instead.
Response: We removed the p-values from the text.
- Figure 4: the authors need to include a proper scale bar in the image with a stated unit of measurement either in the image or the figure description. It would also be helpful if the authors could annotate the images to highlight the areas of necrosis, polymorphous inflammation, and hemorrhage, specifically the fungal elements would be most useful to be pointed out.
Response: We replaced the Figure and added the scale and annotation.
- Figure 5: same comments as Figure 4; the authors need to include a proper scale bar in the image with a stated unit of measurement either in the image or the figure description. Furthermore, the authors should annotate the image to highlight areas of inflammatory exudate and fungal elements.
Response: We replaced the Figure and added the scale and annotation.
- Materials and Methods: the manuscript is completely lacking a methods section describing the histological methods used for the images shown in Figures 4 and 5. The authors need to include this section as it is part of the manuscript.
Response: We included a methods section (2.4.5.) describing the histological methods used for the images shown in Figures 4 and 5.
Thank you again for reviewing our manuscript.
Round 2
Reviewer 2 Report
Comments and Suggestions for Authors
The authors have sufficiently addressed my previous comments/suggestions to improve the scientific merit of their case report. The manuscript reads well and provides sufficient evidence for the study objective.